# MicroRNAs and Long Non-Coding RNAs and Their Hormone-Like Activities in Cancer

**DOI:** 10.3390/cancers11030378

**Published:** 2019-03-17

**Authors:** Barbara Pardini, George A. Calin

**Affiliations:** 1Department of Experimental Therapeutics, The University of Texas MD Anderson Cancer Center, Houston, 1515 Holcombe Boulevard, Unit 422, Houston, TX 77030, USA; 2Department of Medical Sciences, University of Turin, Turin 10126, Italy; 3Italian Institute for Genomic Medicine (IIGM), Turin 10126, Italy; 4Center for RNA Interference and Non-Coding RNAs, The University of Texas MD Anderson Cancer Center, Houston, TX 77030, USA; 5Department of Leukemia, The University of Texas MD Anderson Cancer Center, Houston, TX 77030, USA

**Keywords:** non-coding RNAs, microRNAs, long non-coding RNAs, hormones, hormone-like action

## Abstract

Hormones are messengers circulating in the body that interact with specific receptors on the cell membrane or inside the cells and regulate, at a distal site, the activities of specific target organs. The definition of hormone has evolved in the last years. Hormones are considered in the context of cell–cell communication and mechanisms of cellular signaling. The best-known mechanisms of this kind are chemical receptor-mediated events, the cell–cell direct interactions through synapses, and, more recently, the extracellular vesicle (EV) transfer between cells. Recently, it has been extensively demonstrated that EVs are used as a way of communication between cells and that they are transporters of specific messenger signals including non-coding RNAs (ncRNAs) such as microRNAs (miRNAs) and long non-coding RNAs (lncRNAs). Circulating ncRNAs in body fluids and extracellular fluid compartments may have endocrine hormone-like effects because they can act at a distance from secreting cells with widespread consequences within the recipient cells. Here, we discuss and report examples of the potential role of miRNAs and lncRNAs as mediator for intercellular communication with a hormone-like mechanism in cancer.

## 1. Introduction

The term “hormone” was first introduced in 1905 by Starling, referring to the discovery of secretin [1]. A hormone is a chemical messenger (in general, a peptide or a steroid) produced by the endocrine glands and circulating in the body to regulate the activities of specific target organs at a distal site [2,3]. The mode of action of hormones requires an interaction of these chemical messengers with specific receptors located on the cell membrane or inside the cell. The binding hormone-receptor generates a signaling cascade that modifies cellular activity [4]. The definition of hormones is quite restrictive since not all hormones are originated from endocrine glands, with many of them acting locally via autocrine/paracrine regulation. Specialized cells in various other organs also secrete hormones in response to specific biochemical signals from a wide range of regulatory systems. Serum/calcium concentration, for instance, affects parathyroid hormone synthesis while serum glucose concentration affects insulin synthesis. In addition, since the outputs of the stomach and exocrine pancreas become the input of the small intestine, the small intestine itself secretes hormones to stimulate/inhibit the stomach and pancreas in accordance to how busy it is, in a regulated feedback known more generally as “diffuse endocrine system” [5]. In a broader view, hormones are considered in the cell–cell communication context and in mechanisms of cellular signaling [2,3]. The best-known mechanisms of this kind are chemical receptor-mediated events, the cell–cell direct interactions through synapses, and, more recently, the extracellular vesicle (EV) transfer between cells [6]. EVs are small membrane-enclosed structures produced by different mechanisms that can be secreted from almost all cell types [7,8] in a process evolutionary conserved from bacteria to humans [9]. Each cell type is able to turn on EV biogenesis depending on the physiological states and, also, the EV cargo components can be highly regulated [10]. EVs, such as exosomes and microvesicles, represent the way donor cells communicate with recipient cells and influence their gene expression [11]. In the last years, it has been extensively demonstrated that EVs are used as a way of communication between cells and that they are transporters of specific messenger signals. EVs are, in fact, enriched for specific proteins (as for example cytokines), lipids, messenger RNAs (mRNA), and non-coding RNAs (ncRNAs), such as microRNAs (miRNAs) and long non-coding RNAs (lncRNAs) [6,8,12]. The nature and abundance of EV content are related to the specific cell type, and are influenced by the physiopathological state of the donor cell [13]. The cell–cell communication mediated by RNAs included in EVs has been described for the first time by Valadi et al. in 2007: exosomes carried miRNAs and other RNAs from one cell to another and, when released in the target cell, were able to interact with the gene expression machinery to modify the gene expression profile of the recipient cell [6].

EVs result as an alternative mode of communication between neighboring and distant cells. Respect to conventional mechanisms of cell communication, EVs differ because of specific temporal and spatial properties and mostly because of the potentiality to group multiple signals together [14].

## 2. miRNAs and Their Hormone-Like Activity in Cancer

The advances in high-throughput sequencing technology and bioinformatics have revolutionized ncRNA discovery [15]. Mammalian genomes are highly transcribed and the majority of the transcripts do not code for proteins. However, this high rate of transcription is not done in an indiscriminate way: the cellular repertoire of ncRNAs includes small housekeeping RNAs (such as ribosomal RNAs (rRNAs) and transfer RNAs (tRNAs)), as well as miRNAs and lncRNAs [16].

miRNAs are a class of single-stranded ncRNAs that play a critical role in the negative regulation of gene expression at post-transcriptional level [17]. Thousands of miRNAs have been identified in all eukaryotes and, so far, the latest version of miRBase (release 22 March 2018) accounts for over 38,000 miRNA gene loci in 271 species. In animal cells, miRNAs pair, in a complementary manner, with the 3′UTR of target mRNAs, inhibiting their translation or inducing their degradation [17]. miRNAs are crucial regulators in a wide range of biological processes but they are also implicated in human diseases, including cancer [16,18,19,20]. There are several lines of evidence that miRNAs are involved in endocrinology. It has been demonstrated that miRNAs can regulate directly genes encoding hormones or other enzymes involved in hormone maturation and metabolism. miRNAs can also target hormone antagonists or receptors indirectly modifying the hormone-mediated cell signaling transmission [21,22] or could be regulated by hormones either at the level of miRNA transcription and processing [23,24,25]. For instance, miR-21 and miR-181-b1 genes are expressed after *STAT3* induction, which is activated by interleukin 6 (IL-6) [26]. Moreover, miR-21 is repressed by thyroid hormone (TH) and this downregulation regulates *GRHL3*, a transcriptional inhibitor of type 3 iodothyronine deiodinase (D3) which, in turn, inactivates TH [27].

Recently, the role of miRNAs as mediators for intercellular communication with a hormone-like mechanism has also been established. miRNAs can work as autocrine, paracrine, and endocrine messengers. In fact, the classic mechanism of action of a miRNA is to be transcribed by a cell and induce local signaling on that same cell (autocrine signaling, Figure 1A). On the other hand, a miRNA can also transmit local signaling between nearby cells (paracrine signaling, Figure 1B) [8]. In cancer, the intercellular signaling mediated by miRNAs has been related to the tumor microenvironment (TME) setting or pre-metastatic niche induction [28,29,30]. An example of this kind of signaling in cancer is the one in which tumor-derived EVs containing miRNAs can directly modify tumor cell invasiveness and motility through modification of the TME [31,32]. Moreover, it has been observed that the ectopic expression of miR-409 in normal prostate fibroblasts conferred a cancer-associated stroma-like phenotype. The release of this miRNA via EVs was able to promote tumorigenesis and epithelial-to-mesenchymal transition (EMT) through repression of Ras suppressor 1 (*RUS1*) and stromal antigen 2 (*STAG2*), well-known tumor suppressors [33]. The discovery of miRNAs in extracellular fluids or loaded in EVs, such as exosomes, is the main evidence that miRNAs may act as paracrine and endocrine interactors [11,34]. EVs containing miRNAs can either work locally or distally via transport within the circulatory system. Moreover, miRNAs circulating within bodily fluids and extracellular fluid compartments may have an endocrine hormone-like effect because they can reach cells that are distant from the secreting cell, modifying their gene expression (Figure 1C) [34,35]. Once released, the EVs containing miRNAs can interact with a recipient cell, deliver its cargo to the cytosol, and modulate the phenotype of the target cell [36]. There is evidence that demonstrates that miRNAs in EVs can be taken up into neighboring or distant cells and modulate the function of those recipient cells in many physiological and pathological conditions [37,38,39]. For example, Le et al. demonstrated, in vitro and in vivo, that murine and human metastatic breast cancer cells release miR-200 family miRNAs to nonmetastatic cells via EVs. The transfer of these molecules altered gene expression in the recipient cells (which were lung cancer cells in the in vivo experiments) and promoted mesenchymal-to-epithelial transition [40]. Interestingly, the exosomal miRNA cargo occurs non-randomly and, also, the recipient cells are finely targeted, enforcing the idea of specific function for a single miRNA on a specific target [6,16]. miRNA species that are transported via EVs do not reflect the miRNA expression profiling of the donor cells [41,42] and, interestingly, several studies demonstrated that cancer patients have elevated levels of tumor-derived exosomes in plasma or serum compared with healthy controls [43,44,45]. The secretion of tumor-specific miRNAs via exosomes indicates the importance of this mechanism in influencing the surrounding microenvironment [34].

Functional interactions between cancer cells and the TME are mediated by small molecules such as cytokines and growth factors [46]. In addition, cancer cells may also transfer important functional information through paracrine communication via EVs [47]. EV cargo may actually influence the stroma by activating molecular pathways that differ from those mediated by soluble factors [8]. Therefore, tumor-derived EVs can alter the physiology of surrounding cells and distant non-tumor cells to facilitate cancer dissemination and growth [31].

The TME (which includes extracellular matrix, cancer-associated fibroblasts, tumor-associated macrophages, immune cells, and others) plays a crucial role in all steps of carcinogenesis [48]. Several examples of this biological information transfer between malignant cells and TME components via EV-transported miRNAs have been reviewed in a recent work by Bayraktar et al. [8]. For instance, Baroni et al. [49] found that in cancer-associated fibroblasts from triple-negative breast cancer patients, miR-9 was upregulated when compared with normal fibroblasts. Moreover, miR-9 was released by tumor cells and transferred via exosomes to normal fibroblasts recipients which, as a consequence, overexpressed this miRNA and increased their motility. Therefore, high expression of miR-9 in fibroblasts affects breast cancer progression [49]. Another example was described in the work of Chen et al. in which it was found that miR-940, released in exosomes by ovarian cancer cells, targeted tumor-associated macrophages and promoted tumor growth via the CD206 and CD163 pathways [50].

The assumption that a miRNA might work as a hormone or with a hormone-like mechanism implicates the possibility of the existence of a protein receptor for miRNAs (defined as miRceptor by [34]) and miRNA–protein interaction. The first study demonstrating miRNA–protein binding was published in 2010 by Eiring et al., where the authors provided evidence of steric binding between miR-328 and hnRNPE2 in blast crisis of chronic myelogenous leukemia. This “decoy activity” of miRNA prevents hnRNPE2 binding to *CEBPA* mRNA, thus restoring *C*/*EBPα* expression that further and directly enhances miR-328 transcription [51].

In 2012, it was demonstrated that EVs containing miR-21 and miR-29a released by non-small cell lung cancer cell lines were targeting tumor-associated macrophages and, more specifically, the human toll-like receptor 8 (*TLR8*), triggering the downstream pathway. As a result, authors observed an increased secretion of IL-6 and tumor necrosis factor-α (TNFα) by tumor-associated macrophages, which determines a pro-tumoral inflammatory response promoting cancer growth and metastasis [52].

Patel and Gooderham observed that IL-6 triggers the IL-6R/STAT3 pathway and also increases miR-21 and miR-29b expression in colorectal cancer cells. The authors proposed a model in which these miRNAs are released via exosomes and reach immune cells, where they interact with the TLR8 miRceptor. This interaction may induce an increase of IL-6 in a feed-forward loop involving miRNA–miRceptor interactions which are responsible for the increased secretion of IL-6, a typical phenomenon in the colorectal cancer microenvironment [53]. Interestingly, a similar mechanism has also been found in neuroblastoma. Endovesicular miR-21, released by neuroblastoma cells, binds to *TLR8* in surrounding tumor-associated macrophages, inducing in these cells the upregulation and the release in EVs of miR-155. Macrophage-derived EVs containing miR-155 are transferred back to neuroblastoma cells where miR-155 acts on its target, telomeric repeat-binding factor 1 (*TERF1*, a telomerase inhibitor). The silencing of *TERF1* induces increased resistance to cisplatin in neuroblastoma cells [54].

miRNAs released by exosomes and working in a hormone-like fashion could also be an optimal therapeutic target in the case of tumor drug resistance [55]. Wei et al., for example, demonstrated the role of exosomal miR-221/222 in the resistance to tamoxifen in breast cancer cells [56]. In another study, it was demonstrated that cancer-associated fibroblasts released exosomes containing miR-21, miR-378e, and miR-143-3p, that were able to induce stemness and epithelial–mesenchymal transition phenotypes in breast cancer cell lines [57].

An intriguing aspect that could be bound to the hormone-like action of miRNAs has been raised by Zhang et al. in 2012 [58] and reinforced by Zhou et al. in 2015 [59]. In these works, researchers demonstrated the possibility that miRNAs derived from plants could potentially travel, through food, from plants to animals via the gastrointestinal tract and access host cellular targets, where they work as bioactive compounds able to influence recipients’ physiopathological conditions. The authors proposed that epithelial cells in the intestine could absorb plant-derived miRNAs contained in food and include them into EVs to protect them from degradation and facilitate their release into the blood stream. These “exogenous miRNAs” then seem to be able to reach organs and tissues via circulation and modulate gene expression. The evidence supporting this theory has been summarized in a recent review by Li et al. [60]. This sort of plant–animal communication, named cross-kingdom transmission, is still source of debate in the scientific community. In fact, there is a large amount of evidence contradicting this cross-kingdom communication hypothesis (also widely reviewed by [60]). The main concern is the mechanisms by which exogenous miRNAs can bypass and survive in the gastrointestinal tract, to enter the bloodstream and ultimately reach specific targets. This “exogenous post-transcriptional regulation” could be another factor influencing the development in special cases of diseases, such as cancer, inserting additional levels of complication into an already complicated scenario. If validated, this hypothesis may expand the current knowledge on dietary bioactive compounds and their biological actions once internalized in the organism [60,61].

## 3. Long Non-Coding RNAs Acting as Hormones

miRNAs are the most studied species of ncRNAs but, in the last years, the attention of researchers has also been focused on other ncRNAs whose functions are still not well described. A special mention should be made for lncRNAs, since their biological roles and mechanisms of action are not yet completely understood, especially in the context of carcinogenesis [62]. Assigning molecular, cellular, and physiological functions to lncRNAs is among the greatest challenges of the next decade, and there is now increased attention on their biological functions in hormonal signaling systems [63,64,65,66].

lncRNAs are defined as non-protein coding RNA transcripts larger than 200 nucleotides, but this definition is quite vague since a universal scheme does not exist [62,67]. The working definition for lncRNAs includes all RNA molecules longer than 200 nucleotides, having little coding potential, transcribed by PolII, capped, spliced, and polyadenylated [63]. The expression of lncRNAs is dependent on the cellular, tissue, and metabolic context. As a consequence, there are specific lncRNAs associated with specific cellular processes that may be inferred by their differential pattern of expression in tissues but also in different developmental time points or under specific stimuli [61,63,68]. It is a common belief that lncRNAs are mostly involved in transcriptional regulation and, therefore, reside principally in the nucleus. However, several lncRNAs act, or are even exclusively localized, within the cytoplasm by working as post-transcriptional regulators in interaction with miRNAs, mRNAs, or proteins [69,70,71,72]. Interestingly, the EV cargo may be enriched in lncRNAs [10,73,74], as observed in plasma exosomes of patients with castration-resistant prostate cancer [75] and in renal cancer [76]. The scenario is even more complicated due to a large number of lncRNAs that have been implicated in competing endogenous RNA (ceRNA) mechanisms. This is possible since lncRNAs can function as sponges, able to bind and reduce the targeted effects of miRNAs on mRNAs [77,78].

lncRNA have been recognized as having endocrine, paracrine, and autocrine regulatory functions in a way similar to the one already described for miRNAs [74]. In fact, they can have an autocrine hormone-like behavior since they can modulate cellular activity directly by controlling transcription. For example, they can interact with hormone-encoding genes or hormone antagonists/receptors, indirectly modifying the cell signaling transmission [79]. Steroid receptor RNA activator (*SRA*) was among the first lncRNA to be associated to hormone receptor pathways and acting with a hormone-like mechanism. *SRA* is expressed in tissues specifically targeted by steroid hormone, and it works as a co-activator of the steroid receptor to facilitate ligand-dependent transactivation [80]. Additionally, *SRA* can interact with co-repressors of nuclear receptors [81]. Different expression patterns of *SRA* have been observed in breast cancer cell lines [82], demonstrating that its hormone receptor-associated activity may be crucial in breast tumorigenesis.

Growth arrest-specific 5 (*GAS5*) is another interesting example of multifunctional lncRNAs. It works as a multiple nuclear receptor decoy, forming an RNA stem–loop structure that mimics nuclear receptor DNA response elements. For example, it interacts with glucocorticoid DNA binding domain working as a decoy for glucocorticoid receptor response element [83]. As a consequence, the glucocorticoid receptor is liberated from its sites of transcriptional activity. Therefore, the overexpression of *GAS5* blocks cell growths and induces apoptosis in adherent human cell lines. On the other hand, its reduced expression has been observed in human breast cancer cell lines, indicating a possible involvement of this lncRNA in breast carcinogenesis [84].

lncRNAs can also travel via EVs to nearby or distant cells, where they can induce specific phenotypical changes in a paracrine and endocrine way [31]. The most interesting examples in cancer apply to drug resistance, angiogenesis promotion, and tumorigenesis induction [74].

The ability of tumor cells to disseminate the drug-resistant phenotype via exosomes has been recognized mainly through transferring of miRNAs and drug-efflux pumps [85]. However, there is substantial evidence supporting a role for lncRNAs embedded in exosomes in this mechanism. Expression levels of exosomal lncRNAs are greatly different from those of the donor cells, and there is evidence that lncRNAs are not randomly secreted in EVs [86,87].

The role of EVs and lncRNAs in tumor progression and aggressiveness has been demonstrated in several studies reviewed by Andaloussi et al. [55]. The lncRNA called metastasis-associated lung adenocarcinoma transcript (*MALAT1*) regulates alternative splicing and gene expression [88,89] contributing to lung cancer metastasis [90]. In addition, high levels of *MALAT1* have been detected in serum exosomes from non-small cell lung cancer patients and connected with the promotion of cell proliferation and migration of this cancer [91].

Notably, Qu and collaborators demonstrated, in an elegant way, that lncRNA activated in renal cell carcinoma with sunitinib resistance (*lncARSR*) is correlated with poor response to sunitinib, a drug used for the treatment of advanced renal cell carcinoma. The resistance to the drug was directly induced by *lncARSR* that works as a ceRNA for miR-34/miR-449 to facilitate the expression of specific genes implicated in the sunitinib resistance. Most interestingly, the authors found that *lncARSR* is incorporated into exosomes and transmitted to sensitive cells for the dissemination of the resistance in a hormone-like fashion. The transmission of resistance is not only between tumor cells but also involves endothelial cells, implicating that the exosome-mediated communication is also between tumor and stromal cells [76]. The exosomal secretion of lncRNAs is highly selective and different between normal and cancer cells or between sensitive and resistant cells, therefore, identifying cellular molecules responsible for RNA secretion may help in finding a strategy to block this cell-specific mechanism [76].

Lang and collaborators found that glioma cells were enriched in POU class 3 homeobox 3 (*POUF3*) lncRNA. These cells were able to release *POUF3* into the exosomes and target the surrounding normal tissue, inducing cell proliferation, migration, and angiogenesis in an in vivo model [92].

In the last years, another lncRNA, the colon cancer-associated transcript 2 (*CCAT2*), attracted the attention of researchers because of its dysregulation in cancer [65,66,93,94]. Notably, *CCAT2* has been demonstrated to work in a hormone-like fashion [95]. Our group demonstrated an important role of *CCAT2* in regulating *MYC*, miR-17-5p, and miR-20a [96]. Interestingly, *CCAT2* interacts with these targets through TCF7L2 enhancing the WNT signaling activity. However, it has been demonstrated that *CCAT2* is itself a WNT downstream target. Therefore, in colon cancer, there is a feedback loop mechanism between *MYC*, *WNT*, and *CCAT2* [96]. Moreover, *CCAT2* released in exosomes by glioma cells has also been found to be responsible of angiogenesis induction and apoptosis inhibition in endothelial cells [92]. The pro-angiogenesis phenotype of endothelial cells can be induced also by *H19*, another important lncRNA in carcinogenesis. Conigliaro et al. found that CD90+ liver cancer cells can reprogram endothelial cells by releasing *H19*-enriched exosomes [97].

## 4. Conclusions

In conclusion, there is an increasing interest in circulating ncRNAs as mediators of cell–cell communication and regulators of gene expression in recipient cells. The concept that an ncRNA might function as a hormone (i.e., mediating cells communication) is a challenge for the research community, and the current knowledge is still insufficient for clarifying this topic. Understanding the role of exogenous ncRNAs that could work as messengers in inter-individual and cross-species molecular communication is one of the next scientific targets for researchers. There is high potential for clinical applications not only as diagnostic or prognostic biomarkers but also as therapeutics [98]. Given the rapid and extensive progress made in the field of ncRNAs in the last decade, in the near future, researchers will be able to address these challenges.

## Figures and Tables

**Figure 1 cancers-11-00378-f001:**
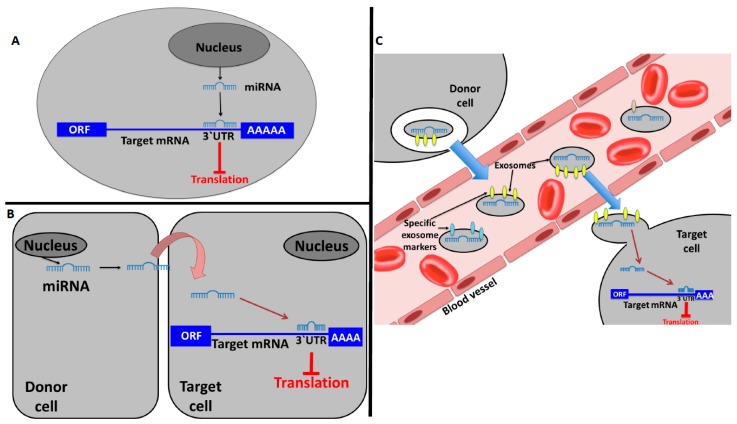
MicroRNAs (miRNAs) working in a hormone-like fashion. (**A**) Autocrine communication: a miRNA produced by a cell binds to autocrine receptors of the same cell inducing a local signaling. (**B**) Paracrine communication: a miRNA produced by a cell transmit a local signaling between nearby cells. (**C**) Endocrine communication: an extracellular vesicle (EV)-embedded miRNA is the mediator of distant signaling.

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
