# Peer review of "MicroRNAs and Long Non-Coding RNAs and Their Hormone-Like Activities in Cancer"

_cancers, 2019, doi:10.3390/cancers11030378_

Round 1

Reviewer 1 Report

In the review “Non coding RNAs and their hormone-like activities in cancer”, the authors describe ncRNA functions in the wider context of signaling molecules that can be considered as hormones, due to their property of acting on recipient cells at a distal site, or to interfere with hormone-mediated signaling pathways. These concepts are new and intriguing, and could allow to put many information in a different context and open new perspectives.

The specific articles collected are pertinent and sufficiently up to date. However, it would be advisable to cite more primary literature (research articles) to support the concepts proposed, instead of referring to other reviews, especially if from the author(s) of this manuscript (e.g. at line 82, 91, 93, 162…).

Line 40: “not all the hormones are originated from endocrine glands”. Please mention some examples.

Line 42: To help the reader conceptualize the message the authors want to convey about the modern vision of hormones, some more literature should be cited. Ref [2] cited in this position doesn’t support the concept.

Line 63: Please specify that this is related to the proposed autocrine mode of action, as showed in figure 1A.

At line 82: “several studies demonstrated that cancer patients have elevated levels of tumor-derived exosomes in plasma compared with healthy controls”. Ref [30] conclusions are that serum levels of tumor-derived miR-141 (and not exosomes) is higher compared to healthy controls. Please rephrase accordingly, or include different articles. There is a considerable literature regarding different expression patterns of miR, carried by EVs, in cancer patients vs healthy controls.

Ref [38] at line 118: in this article, the interaction between miR-21 or miR-29b and TLR8 miRceptor is rather a suggestion than an observation of experimentally validated data. Please rephrase accordingly.

At line 186, the reference about GAS5 function related to the glucocorticoid receptor activity is missing.

Refs [73,74] do not refer to the contribution of MALAT1 to lung cancer metastasis.

Author Response

Reviewer 1

In the review “Non coding RNAs and their hormone-like activities in cancer”, the authors describe ncRNA functions in the wider context of signaling molecules that can be considered as hormones, due to their property of acting on recipient cells at a distal site, or to interfere with hormone-mediated signaling pathways. These concepts are new and intriguing, and could allow to put many information in a different context and open new perspectives.

The specific articles collected are pertinent and sufficiently up to date. However, it would be advisable to cite more primary literature (research articles) to support the concepts proposed, instead of referring to other reviews, especially if from the author(s) of this manuscript (e.g. at line 82, 91, 93, 162…).

Reply:

Thanks a lot for nice comments and suggestions. We reviewed the text and updated the references.

Line 40: “not all the hormones are originated from endocrine glands”. Please mention some examples.

Reply:

Hormones are mainly secreted by endocrine glands. However, specialized cells in various other organs also secrete hormones. Hormone secretion occurs in response to specific biochemical signals from a wide range of regulatory systems. As an example, serum/calcium concentration affects parathyroid hormone synthesis; blood sugar (serum glucose concentration) affects insulin synthesis. In addition, since the outputs of the stomach and exocrine pancreas (the amounts of gastric juice and pancreatic juice) become the input of the small intestine, the small intestine itself secretes hormones to stimulate/inhibit the stomach and pancreas based on how busy it is.

Line 42: To help the reader conceptualize the message the authors want to convey about the modern vision of hormones, some more literature should be cited. Ref [2] cited in this position doesn’t support the concept.

Reply:

We reviewed the section and updated the references.

Line 63: Please specify that this is related to the proposed autocrine mode of action, as showed in figure 1A.

We thank the Reviewer for his/her comments and suggestions. We modified the text of the figure and also expanded in the text the hormone-like mechanism applied to microRNAs.

At line 82: “several studies demonstrated that cancer patients have elevated levels of tumor-derived exosomes in plasma compared with healthy controls”. Ref [30] conclusions are that serum levels of tumor-derived miR-141 (and not exosomes) is higher compared to healthy controls. Please rephrase accordingly, or include different articles. There is a considerable literature regarding different expression patterns of miR, carried by EVs, in cancer patients vs healthy controls.

Reply:

We revised the whole section and updated the literature accordingly.

Ref [38] at line 118: in this article, the interaction between miR-21 or miR-29b and TLR8 miRceptor is rather a suggestion than an observation of experimentally validated data. Please rephrase accordingly.

Reply:

Also another Reviewer asked for clarification. We added more examples and updated the literature accordingly.

At line 186, the reference about GAS5 function related to the glucocorticoid receptor activity is missing.

Reply:

We apologize for this. We added the reference

Refs [73,74] do not refer to the contribution of MALAT1 to lung cancer metastasis.

Reply:

Thanks for your comment. We updated the section with the proper reference

Reviewer 2 Report

The concept of non-coding RNA molecules acting as hormones is very interesting and current. The review concentrates on two classes of non-coding molecules - microRNA and long non-coding RNA molecules - perhaps a change of the title would be in order?

However, I believe that an extensive correction of the paper is necessary as there are many basic grammatical mistakes (incorrect word order, incorrect words etc) making it hard to follow. Gene symbols are not always put into italics. Moreover, there are some factual mistakes (perhaps missing words) - p. 2, line 44- EVs are small membrane produced by....no, EVs are not just membranes...

I find the paper hard to follow at this point.

p.1 abstract - 'hormones are chemical messenger' please change to 'messengers'

p1 line 41 - 'in a more modern vision'...????

p.2 - line 73 - 'it has been established also the role' - please correct word order

p.2 line 74 - 'way of action' - please correct

p.2 line 74 'the discovery of miRNAs into extracellular fluids' - do the authors mean discovery IN extracellular fluids?

p3 line 102 - 'miR-940 relased ...were targeting' - who was targeting? the miRNA or the cells? not clear

p3 line 107 - 'decoy a ctivity' - please correct activity

p3 line 112  - please add abbreviation for tumor necrosis factor A

the authors use 'an hormone-like'..please correct to 'a hormone-like' in the entire manuscript

the authors also use cell-cell communication written in 3 different ways...please unify.

p4 - line 127 - 'diet/plant' - not sure about the meaning, please, clarify

p 4 line 128 - 'work as a bioactive constituents'  - please delete a, please clarify 'constituents'

p4 line 134 - not sure that a discussion may be 'acute'

p4 line 145 - 'is not done in an indiscriminately way' - indiscriminate way?

p 4 line 147 - please correct 'ribosomial' into 'ribosomal'

p4 line 148 - 'for lncRNAs there is a large interest in clarifying better their' ....word order not correct

p4 line 155 - please correct 'polyadenilated' to 'polyadenylated'

p4 line 156 "dependent to the cellular"...please correct

p5 line 174 - please clarify ' as for miRNAs, lncRNAs can regulate'

p 5 line 182-183 - please delete 'a' in 'that mimics a nuclear receptor DNA response elements'...

p5 line 194 - please correct 'the role of exosomes...have been' to 'has been'...

p5 line 199 please correct 'regulate' to 'regulates'

p5 line 204 - 'drug employed' please correct, wrong expression

p6 line 216 - please clarify statement "mediating cells communication in an individual'

p6 line 217 - 'more challenging is also to clarify" - please correct

p6 line 221-22 - progresses - not usually used in plural

p6 line 221 -'high chances'

Author Response

Reviewer 2

The concept of non-coding RNA molecules acting as hormones is very interesting and current. The review concentrates on two classes of non-coding molecules - microRNA and long non-coding RNA molecules - perhaps a change of the title would be in order?

Reply:

We thank the Reviewer for his comments and suggestions. We modified the title accordingly.

However, I believe that an extensive correction of the paper is necessary as there are many basic grammatical mistakes (incorrect word order, incorrect words etc) making it hard to follow. Gene symbols are not always put into italics. Moreover, there are some factual mistakes (perhaps missing words) - p. 2, line 44- EVs are small membrane produced by....no, EVs are not just membranes...

Reply:

We apologize for the mistakes and incorrect words. We review completely the manuscript and checked for spelling and grammar errors. We also modified the gene symbols and corrected the sentence highlighted by the Reviewer.

I find the paper hard to follow at this point. 

p.1 abstract - 'hormones are chemical messenger' please change to 'messengers'

p1 line 41 - 'in a more modern vision'...????

p.2 - line 73 - 'it has been established also the role' - please correct word order

p.2 line 74 - 'way of action' - please correct

p.2 line 74 'the discovery of miRNAs into extracellular fluids' - do the authors mean discovery IN extracellular fluids?

p3 line 102 - 'miR-940 relased ...were targeting' - who was targeting? the miRNA or the cells? not clear

p3 line 107 - 'decoy a ctivity' - please correct activity

p3 line 112  - please add abbreviation for tumor necrosis factor A

the authors use 'an hormone-like'..please correct to 'a hormone-like' in the entire manuscript

the authors also use cell-cell communication written in 3 different ways...please unify.

p4 - line 127 - 'diet/plant' - not sure about the meaning, please, clarify

p 4 line 128 - 'work as a bioactive constituents'  - please delete a, please clarify 'constituents'

p4 line 134 - not sure that a discussion may be 'acute'

p4 line 145 - 'is not done in an indiscriminately way' - indiscriminate way?

p 4 line 147 - please correct 'ribosomial' into 'ribosomal'

p4 line 148 - 'for lncRNAs there is a large interest in clarifying better their' ....word order not correct

p4 line 155 - please correct 'polyadenilated' to 'polyadenylated'

p4 line 156 "dependent to the cellular"...please correct

p5 line 174 - please clarify ' as for miRNAs, lncRNAs can regulate'

p 5 line 182-183 - please delete 'a' in 'that mimics a nuclear receptor DNA response elements'...

p5 line 194 - please correct 'the role of exosomes...have been' to 'has been'...

p5 line 199 please correct 'regulate' to 'regulates'

p5 line 204 - 'drug employed' please correct, wrong expression

p6 line 216 - please clarify statement "mediating cells communication in an individual'

p6 line 217 - 'more challenging is also to clarify" - please correct

p6 line 221-22 - progresses - not usually used in plural

p6 line 221 -'high chances'

Reply for all the previous comments:

We apologize for the mistakes and incorrect parts. We have modified all the sentences highlighted by the Reviewer and rephrased the sentences when requested.

Reviewer 3 Report

The authors highlight the hormone-like activities of miRNAs. Indeed, miRNA may have activities in common with hormones, such as their role in cell-to-cell communication in an autocrine, paracrine and endocrine fashion. After reading the title, I recommend to make a point by point comparison in figure 1. Not all aspects of hormone actions were discussed and some of the comparisons seem to be stretched. Please find my comments:

-                      Hormones play a physical role in the organism, whereas miRNA present in bloodstream exosomes seem to be involved in cancer. Are there examples of circulating miRNAs with specific physiological roles, or is their hormone-like action restricted to cancer?

-                      Hormones are biosynthesis in a dedicated organ or endocrine tissue. Is there evidence that miRNAs classes are specifically secreted by certain type of cells, besides cancer cells?

-                      It is discussed that miR-328 binds to hnRNP E2 and prevents CEBPA mRNA binding. Could the authors discuss the specificity of this action, e.g. are other hnRNP E2-bound mRNAs also affected by other miRNAs?

-                      Hormones often are part of regulatory loops. A feed forward loop in tumor-associated macrophages for miR-21/miR29b with IL6 signaling is discussed, but miR-21 is also involved in PTEN/Akt and TGFbeta pathway. Is there a specificity in the hormone-like action of this miRNA?

-                      The fact that miRNAs interact with intracellular proteins does not mean that those proteins function as miRceptors. Only two miRceptors examples are given, of which Toll like Receptor 8 binds both miR-21 and miR29a, which raises the question if this is a ligand specific receptor?

-                      The first paragraph of section 2 discusses the interference of miRNAs with hormonal metabolism, rather than being hormone-like. This is of interest, but should be discussed in a separated section.

-                      Exosome content is non-randomly and may be targeted to specific tissues. However, as discussed in section 3, exosomes contain a mixture of miRNAs together with other types of ncRNAs including lncRNAs that may act as miRNA decoys. The authors should discuss to what extend a single miRNA or rather the whole exosome should be considered as the hormone-like mechanism?

Author Response

Reviewer 3

The authors highlight the hormone-like activities of miRNAs. Indeed, miRNA may have activities in common with hormones, such as their role in cell-to-cell communication in an autocrine, paracrine and endocrine fashion. After reading the title, I recommend to make a point by point comparison in figure 1. Not all aspects of hormone actions were discussed and some of the comparisons seem to be stretched.

Reply:

We thank the Reviewer for his/her comments and suggestions. We modified the text of the figure and also expanded in the text the hormone-like mechanism applied to microRNAs.

Please find my comments:

-                      Hormones play a physical role in the organism, whereas miRNA present in bloodstream exosomes seem to be involved in cancer. Are there examples of circulating miRNAs with specific physiological roles, or is their hormone-like action restricted to cancer?

Reply:

miRNAs in the bloodstream and contained in EVs are not only associated with cancer and other diseases (for example metabolic diseases as described in Huang-Doran et al, Trends in Endocrinology and Metabolism, 2017) but represent active and important mediators of cell-cell communication in the organism not only in concomitance of a disease. They are ubiquitous and stable in human biofluids therefore it is more the alterations in the miRNAs cargo in EVs that is associated to diseases like cancer. We were not stressing so much this aspect because the review was focused on the hormone-like functions of non-coding RNAs in cancer. However, we added a sentence to better clarify this aspect.

There are several examples of miRNAs transported in EVs that works as hormones also in non-cancer conditions and, more importantly, the mechanism of action is not only through the bloodstream. An interesting work is the one published by Ying et al, Cell 2017 in which authors demonstrated that exosomes released by adipose tissue macrophages in obese animals were enriched in miR-155. This miRNA was transferred to insulin target cell types via paracrine/endocrine regulation modulating insulin sensitivity (the paper has been discussed in the review).

Moreover, the release of EVs enriched in miRNAs have been observed also in all the available human bodily fluids (as released by the International Society for Extracellular Vesicles, there are more than 30 types of biofluids containing EVs in mammals).

-                      Hormones are biosynthesis in a dedicated organ or endocrine tissue. Is there evidence that miRNAs classes are specifically secreted by certain type of cells, besides cancer cells?

Reply:

It should be stressed that we used the term hormone-like mechanism of action for miRNAs because for these molecules there is no such a high cell-specificity such as the one observed for hormones. With just few exceptions, usually miRNAs has hundreds of possible targets and a single miRNA can be produced by different types of cells. What it is interesting is that the release of miRNAs for cell-cell communication is nonrandom and highly regulated. Only a specific pattern of miRNAs can be retrieved in exosomes released by certain type of cells and this is resembling more similar to the way of action of hormones. There are several examples of miRNAs transported in EVs that works as hormones also in non-cancer conditions and, more importantly, the mechanism of action is not only through the bloodstream. An interesting work is the one published by Ying et al, Cell 2017 in which authors demonstrated that exosomes released by adipose tissue macrophages in obese animals were enriched in miR-155. This miRNA was transferred to insulin target cell types via paracrine/endocrine regulation modulating insulin sensitivity (the paper has been discussed in the review).

-                      It is discussed that miR-328 binds to hnRNP E2 and prevents CEBPA mRNA binding. Could the authors discuss the specificity of this action, e.g. are other hnRNP E2-bound mRNAs also affected by other miRNAs?

Reply:

miR-328 was shown to specifically interact in a seed sequence-independent manner with hnRNP E2 by Eiring et al. This resulted in a missing binding of CEBPA mRNA to hnRNP E2 that rescued CEBPA mRNA translation both in vitro and in vivo (which is usually lost in chronic myeloid leukemia). In other words, miR-328 may compete with CEBPA mRNA for binding to hnRNP E2 that, in turn, releases CEBPA and allows its loading onto poly-somes for translation

MicroRNA interaction with sequence-specific RNA binding protein is not unew; however, the case of hnRNP E2 and miR-328 is particular. For instance, hnRNP A1 is another RNA binding protein upregulated in CML-BC and it is known to binds the primary miR-17-92 transcript to allow processing of pre-miR-18a. The expression of the miR-17-92 cluster is upregulated in CML and is important for proliferation and reduced susceptibility to apoptosis of K562 cells. The binding between miR-328 and hnRNP E2 is not affecting miRNA biogenesis. Moreover, authors demonstrated that there are no binding site of this miRNA with CEBPA therefore the hnRNP E2 binding to miR-328 and CEBPA mRNA are mutually exclusive.

Hormones often are part of regulatory loops. A feed forward loop in tumor-associated macrophages for miR-21/miR29b with IL6 signaling is discussed, but miR-21 is also involved in PTEN/Akt and TGFbeta pathway. Is there a specificity in the hormone-like action of this miRNA?

Reply:

There are several studies that investigated the role of miR-21 in cancer and its involvement in PTEN/Akt and TGFbeta pathway (Li et al 2016 for gastric cancer; Liu et al 2016 in human hepatocytes and Liu et al 2016 in fibroblasts). To the best of our knowledge none of the study currently available investigated the molecular mechanism of this interaction. On the base of the data available we could hypothesize an autocrine/paracrine regulation but there are no data available at the moment verifying this hypothesis.

-                      The fact that miRNAs interact with intracellular proteins does not mean that those proteins function as miRceptors. Only two miRceptors examples are given, of which Toll like Receptor 8 binds both miR-21 and miR29a, which raises the question if this is a ligand specific receptor?

Reply:

The miRceptors topic has been amply reviewed by Muller Fabbri (2017, MicroRNAs and miRceptors: a new mechanism of action for intercellular communication. Phil. Trans. R.Soc. b 373: 2016.0486). The research is still at the beginning and the validated studies are still few. We added some more examples in the section. The specificity of the ligand for miR-21 and miR-29 in the case of Toll-like receptor 8 has been elegantly demonstrated by Fabbri and colleagues (Fabbri et al, Proc. Natl. Acad. Sci. 2012).

-                      The first paragraph of section 2 discusses the interference of miRNAs with hormonal metabolism, rather than being hormone-like. This is of interest, but should be discussed in a separated section.

Reply:

We agree with the Reviewer that this part is a bit out of the scope of the review. We cut down this part and implemented with examples more related to the hormone-like mechanism.

-                      Exosome content is non-randomly and may be targeted to specific tissues. However, as discussed in section 3, exosomes contain a mixture of miRNAs together with other types of ncRNAs including lncRNAs that may act as miRNA decoys. The authors should discuss to what extend a single miRNA or rather the whole exosome should be considered as the hormone-like mechanism?

Reply:

The argumentation raised by the Reviewer is still in development. Much progress have been made in the last years in understanding the biology of EVs and their role in physiological and pathological contexts. The fact that EVs carry tumor-associated or tumor-inducing molecules such as ncRNAs has opened new lines of research for clinical application. However, as observed recently by Tkach and Thery (Cell 2016), the direct demonstration that functional EV-mediated miRNA transfer is the relevant mechanism of cell-cell mechanism is still challenging

Reviewer 4 Report

In the review by Pardini et al. the authors summarize the recent advances in the field of circulating lncRNAs, that is non-coding transcripts that are frequently secreted from tumor cells inside extracellular vesicles. Since lncRNAs are well-known regulators of various cellular processes (and pathologies), the authors summarize the function of these RNA species within the frame of their hormonal-like circulating nature and as mediators of cell-to-cell signaling with functional implications for disease progression. Overall it is a well written review from a respected lab with considerable contribution on the field of circulating ncRNAs. However, I do miss recent examples of circulating lncRNAs in the text. In this particular section of their manuscript, the authors cite approximately 22 papers but only 3 are published between 2015 and 2017. I would therefore like to encourage the authors to significantly revise this part of the manuscript, including recent examples of circulating lncRNAs, emphasizing on those that have been functionally dissected. In addition, the authors should go through their text and correct minor spelling mistakes indicated below prior to acceptance.

Line 20 and 41 change contest to context

Line 73 please rephrase beginning of sentence

Line 145 change “are not coding” to “do not code”

Line 168 change “also lncRNAs can have” to “lncRNAs can also have”

Author Response

Reviewer 4

In the review by Pardini et al. the authors summarize the recent advances in the field of circulating lncRNAs, that is non-coding transcripts that are frequently secreted from tumor cells inside extracellular vesicles. Since lncRNAs are well-known regulators of various cellular processes (and pathologies), the authors summarize the function of these RNA species within the frame of their hormonal-like circulating nature and as mediators of cell-to-cell signaling with functional implications for disease progression. Overall it is a well written review from a respected lab with considerable contribution on the field of circulating ncRNAs. However, I do miss recent examples of circulating lncRNAs in the text. In this particular section of their manuscript, the authors cite approximately 22 papers but only 3 are published between 2015 and 2017. I would therefore like to encourage the authors to significantly revise this part of the manuscript, including recent examples of circulating lncRNAs, emphasizing on those that have been functionally dissected. In addition, the authors should go through their text and correct minor spelling mistakes indicated below prior to acceptance.

Reply:

Thanks a lot for nice comments and suggestions. We reviewed the section of lncRNAs and updated the references.

Line 20 and 41 change contest to context

Line 73 please rephrase beginning of sentence

Line 145 change “are not coding” to “do not code”

Line 168 change “also lncRNAs can have” to “lncRNAs can also have”

Reply for all the previous comments:

We apologize for the mistakes and incorrect parts. We have modified all the sentences highlighted by the Reviewer and rephrased the sentences when requested.

Round 2

Reviewer 1 Report

I checked this last version of the manuscript, together with author’s reply. All the concerns have now being addressed properly, therefore the manuscript is acceptable for publication.

Reviewer 2 Report

Accept in present form. I have nothing else to add.

Reviewer 4 Report

The authors corrected all relevant part of the manuscript therefore now it is suitable for publication.